# Comparing Foundation Models for Medical Images: A Study on Limited Data and Generalization

Ingrid Utseth[*1], Amund H. Vedal[*1], Sarina Thomas[2], and Line Eikvil[**1]

[1]Norwegian Computing Center
[2]GE HealthCare

## Abstract

In this study we investigated how vision foundation models, pretrained on different domains, compete with a specialized model for classification as a function of the size of the labeled training set of medical images. Furthermore, we looked into the different models' ability to generalize to difficult cases.

Our experiments are conducted for cardiac ultrasound images and the downstream task of view recognition. Still, this classification task is meant to serve as a demonstrative example, where we think that the findings should be transferable to other classification tasks and other domains.

Through these experiments we found that the foundation models were able to beat the performance of our task-specific supervised model when labelled training data were limited. This was true even for models trained on natural images and when using the simple linear probing method to create a classifier. We observed that more domain-specific foundation models achieved an even higher performance with limited data. On the other hand, the more general models showed a greater ability to generalize and perform well on difficult, out-of-distribution cases. Still, for typical in-domain cases with sufficient labeled data, a task-specific ResNet model was competitive with the foundation models, while also being both smaller and faster.

## 1 Introduction

The scarcity of high-quality, labeled data is a significant challenge in medical imaging, often hindering the development of robust machine learning models. Traditional supervised learning methods, which require large annotated datasets to achieve high performance, are frequently impractical in this domain. This limitation highlights the need for alternative approaches that can leverage models trained on vast amounts of data to perform well with limited specialized annotations.

Foundation models, which are trained on large, diverse datasets in a self-supervised or unsupervised manner, have demonstrated remarkable success in fields like natural language processing and computer vision. These models learn rich, general-purpose feature representations from their pre-training data, allowing them to be adapted for a wide range of downstream tasks with minimal fine-tuning. However, their effectiveness in specialized domains like medical imaging, and specifically their ability to generalize from natural images to complex medical data, remains a critical area of investigation.

This study investigates the performance of pretrained foundation models against a specialized, supervised model in the context of cardiac ultrasound imaging. We use view recognition, a fundamental classification task, as a demonstrative example to assess model performance as a function of the size of the labeled training set. While our experiments are focused on this specific domain, we think our findings on model performance and generalization may be of interest also for other tasks and domains.

Our main aim in this study is to determine if foundation models, even those pre-trained on natural images, could achieve classification performance comparable to or superior to a specialized supervised model when trained with limited labeled medical data. We also aim to investigate how these general foundation models perform against more domain-specific ones. Finally, we want to analyze their ability to generalize to difficult and ambiguous cases and compare this against a specialized model trained on the same data.

## 2 Related work

With the growing interest in using foundation models for medical imaging, recent studies have investigated their performance and other aspects in comparison to both domain-specific foundation models and task specific models.

Huix et al. [1] tested the performance of five general foundation models trained on natural images on downstream tasks for four medical image datasets under different adaptation schemes. They found that most models underperformed, but DINOv2 performed well with minimal fine-tuning. Cekmeceli et al. [2] have focused specifically on medical image segmentation investigating whether foundation models enhance domain generalization. In their ex-

---

*Equal contribution

**Corresponding Author: eikvil@nr.no

Proceedings of the 7th Northern Lights Deep Learning Conference (NLDL), PMLR 307, 2026.

periments various anatomies and modalities encompass the different domains, where foundation models show a better performance across domains than a Unet segmentation. Performance across domains is still very far from the in-domain performance, and for the in-domain experiments Unet competes well with the foundation models. A study by Chao et al. [3] is even more focused, investigating the performance of foundation models versus domain-specific models for left ventricular segmentation on cardiac ultrasound. They found that a fine-tuned SAM model outperformed EchoNet and U-Net models and demonstrated strong generalization.

In our study we want to investigate more specifically how the models perform for varying sizes of labeled training sets and also compare different types of models, both general and domain specific, where all the foundation models that are investigated are easily available for everyone. Furthermore, we want to compare the different models' ability to generalize to more difficult out-of-distribution cases.

# 3 Methods

## 3.1 Pretrained foundation models

Pretrained foundation models are models that have been trained using self-supervised learning (SSL), a method that trains a model to perform a pretext task – like reconstructing missing regions or aligning different augmented views – thereby eliminating the need for labels required in standard supervised learning.

Several pretrained foundation models for vision exist and are available as open-source models. We have investigated a selection of general state-of-the-art and more specialized open-source pretrained foundation models for vision. Our analysis focused on well-established, easily accessible models on HuggingFace, drawing from general-purpose, general medical, and specialized echo-cardiac models for a broad comparison.

We selected three well-established general models: ViTMAE [4] and DINOv2 [5], which are pure vision models trained on standard RGB images, and CLIP [6], a multimodal text-image model trained on captioned internet images. We also chose two models more specialized for the medical field: MedImageInsight [7], trained on images and text from various medical domains, and EchoCLIP [8], a model focused on cardiac ultrasound that was trained on pairs of cardiac ultrasound videos and reports. In addition to being state-of-the-art and easily available these models are also in general well-documented and well-proven and have robust and well-designed codebases that make them easier to implement and adapt compared to many specialized, ad-hoc architectures. Other specialized medical models exist [9],

[10] [11], but were not selected because they were not available on HuggingFace, were not sufficiently general or sufficiently specific or were not directly suitable for downstream classification tasks.

### 3.1.1 General-purpose models

The **ViTMAE** (Vision Transformer Masked Autoencoder) model is built based on a self-supervised learning (SSL) approach that relies on an asymmetric encoder-decoder architecture. During pretraining a high portion (75%) of the image patches is randomly masked out. Then the encoder is used to encode the unmasked patches. Next, a learnable (shared) mask token is added at the positions of the masked patches. The decoder takes the encoded visual patches and mask tokens as input and reconstructs raw pixel values for the masked positions. By pre-training the model, it learns an inner representation of images that can then be used to extract features useful for downstream tasks. The model is trained on ImageNet.

**DINOv2** (self-DIstillation with NO labels) is a self-supervised learning model that employs a self-distillation technique. This model has been trained on a curated dataset consisting of around 140 million images from a publicly available repository of crawled web data. During training, two variations of an image are passed through two separate networks with identical architecture: a student (which receives a local view) and a teacher (which sees the whole image). Both the student and teacher are Vision Transformers (ViT) models. The teacher is the exponential moving average of the student weights, rather than being updated with backpropagation. The student network is trained to mimic the output of the teacher network.

**CLIP** (Contrastive Language-Image Pre-training) is a multimodal model trained to map text-image pairs to the same embedding space. The model has been trained in a self-supervised way from 400 million text-image pairs publicly available on the internet. It has two main components, a text encoder (which embeds the text) and an image encoder (which embeds the images). For the text encoder a Transformer is used. For the image encoder a few variants have been used, where the Hugging Face variety uses the Vision Transformer. The encoders are trained to maximize the similarity of (image, text) pairs via a contrastive loss. The CLIP model has been shown to have good zero-shot capabilities on various computer vision tasks.

### 3.1.2 General Medical model

**MedImageInsight** is an open-source medical imaging embedding model which is built on a contrastive learning framework, similar to CLIP with both an image and a text encoder. The image encoder uses

a Dual Attention Vision Transforme (DaViT) [12] and is trained using Unified Contrastive Learning (UniCL) [13]. The model is trained on around 3.8M medical images with associated text and labels across a diverse collection of domains, including X-Ray, CT, MRI, dermoscopy, OCT, fundus photography, ultrasound, histopathology, and mammography.

### 3.1.3 Echocardiac model

The **EchoCLIP** model is built on the CLIP framework, but while the original CLIP model is trained on general image-text pairs from the Internet, this model is trained specifically on cardiac ultrasound data. An image encoder and a text encoder have been trained on one million pairs of sampled images from echo videos and EHRs (electronic health records). All the videos show the transthoracic four chamber (4CH) view, a standard view acquired during diagnostic echocardiography. The training has followed the approach of the original CLIP paper with a few minor tweaks., e.g. the image encoder is a ConvNeXt architecture.

## 3.2 Task-specific supervised model

### 3.2.1 ResNet50

The task-specific model used for comparison with the foundation models, is a CNN-model based on the ResNet architecture that we trained supervised specifically for our downstream classification task.

## 3.3 Overview of model characteristics

Table 1 gives an overview of the type and dimensions of the different models we have studied. The MedImageInsight model has a very large number of parameters compared to all the other models, while our specialized model has the lowest number of parameters and the highest embedding dimensions.

**Table 1.** Overview of the models, including the embedding dimension and number of parameters for their image encoder and the size of the pretraining dataset.

| Model | Type | Emb. | #Params | #Pretrain |
|---|---|---|---|---|
| ViTMAE | Img | 768 | 85.8M | 1.2M |
| DINOv2 | Img | 768 | 86.6M | 140M |
| CLIP | Img+Txt | 512 | 87.5M | 400M |
| EchoCLIP | Img+Txt | 512 | 88.1M | 1M |
| MedImageInsight | Img+Txt | 768 | 360M | 3.8M |
| ResNet50 | Img | 2048 | 25.6M | |

## 3.4 Adaption to downstream tasks

Adapting foundation models to downstream tasks typically involves fine-tuning the model's parameters using a smaller, task-specific dataset. The methods for this range from keeping the underlying foundation model untouched, to modifying only a few parameters or updating the entire model.

Our investigation explores two common methods that represent opposite ends of this spectrum: linear probing and full fine-tuning. *Linear probing* is a computationally efficient method that only modifies a final classification layer, whereas *full fine-tuning* updates all parameters for a more thorough, but costly, adaptation.

### 3.4.1 Linear probing

Linear probing is a common, lightweight method to evaluate a pretrained model on a downstream classification task. It involves freezing all parameters of the pretrained model, and training only a linear classification on top of the frozen model. In our case we do this by training a linear support vector machine (SVM) model on features extracted by the pre-trained foundation models, without changing the pre-trained model. Probing the foundation models in this way allows us to evaluate their performance "out-of-the-box" on the ultrasound data. The approach is also computationally efficient, as it requires only a small number of parameters to be trained. However, its performance may be limited if the target task is very different from the original pre-training task, or when the features are not linearly separable.

### 3.4.2 Full finetuning

Linear probing has been a common and simple way to compare and evaluate performance, but in 2022 He et al. [4] re-introduced fine-tuning as the main evaluation metrics. Their main arguments were that linear-probing performances were uncorrelated with that of fine-tuning, and that simple heads do not evaluate the strength of the method to create strong but non-linear features.

For our experiments we have therefore also included finetuning, where we do full finetuning to fully leverage the models' capabilities. With this approach, all of the model's parameters are unfrozen and updated during training on labelled data from the target task. This method often achieves better performance as the model may develop better feature representations for the target task during the training. However, this method is more computationally heavy than linear probing and also requires that suitable hyperparameters are found for the finetuning, e.g. a lower learning rate than training from scratch is usually required.

# 4 Experiments

The aim of the experiments has been to investigate how the different models perform as a function of the size of the training set as well as the different models' ability to generalize, where this is demonstrated for the downstream task of view recognition.

## 4.1 Downstream task

View recognition in cardiac ultrasound is the task of automatically identifying the specific anatomical view of the heart from an ultrasound image or video clip. This is a critical step in the diagnostic workflow as the heart's complex 3D structure is visualized through multiple 2D planes, and each view provides unique diagnostic information. The ASE (American society of echocardiography) [14] published guidelines to suggest several standard views in diagnostic TTE (transthoracic echocardiography) that cover different probe positions and view port poses. Having an automated solution for identifying those standard views is helpful to improve the cardiologists' workflow efficiency, and many such solutions based on deep neural networks exist.

In our experiments we have chosen to use this as a downstream task to investigate and compare the performance of different types of models for varying amounts of training data. This task was chosen as larger amounts of training data are more easily available than for tasks requiring more difficult and time-consuming annotation needed e.g. for specific diagnoses. Furthermore, this is a case where specialized models generally perform well already. For that reason, it is interesting to investigate whether and when foundation models may still have an advantage for example in limited data settings.

## 4.2 Dataset

The dataset used for our experiments consisted of scan-converted cardiac ultrasound sequences from 10 different view classes: apical two chamber (2CH), apical four chamber (4CH), apical five chamber (A-5CH), apical four chamber view with focus on the right ventricle (A-RV), apical three chamber view (APLAX), view without the heart visible (NO-ORGAN), parasternal short axis view (P-SAX), parasternal long axis view (PLAX), parasternal short axis view with aortic valve (SAX-AV) and subcostal probe position (SUBCOSTAL).

Our main dataset came from four different hospitals, where images from three of the hospitals were reserved for training while images from the fourth hospital were used for testing (T1). The total training set consisted of around 7000 sequences ( 700 000 frames). Each sequence covers at least one full heart cycle with the original full time-resolution. The number of frames per sequence varies considerably between patients. On average, there are around 100 frames per sequence. In addition, comes four augmented versions of each image in the full training dataset (2 images augmented with tilt and 2 images augmented with width variations). From the original images, creating nearly 2.9 million images for training. All the images had a size of 256x256 pixels.

The T1 test set consisted of around 1000 sequences ( 5000 frames), where five frames had been selected at intervals from one heart cycle of each sequence. This test set represents a cross-section of examinations typically performed in a hospital.

In addition to this test set, we also included a second dataset (T2) from a separate external clinical site, for which particularly difficult cases were collected representing a range of abnormal anatomies, pathologies and low image quality where correct view categorization can be difficult. Due to this specific selection, this set is much smaller (320 images) than T1 and not all 10 classes are included. Similarly to T1, five frames were selected from each of the sequences. The datasets were all prepared and curated by clinical experts.

The classes, along with the respective sizes of their training and test sets, are summarized in Table 2. The class imbalance observed in this data mirrors the typical distribution of views encountered in clinical practice during cardiac ultrasound examinations.

**Table 2.** Overview of training- and testsets

| Class | Training set | | Test T1 | | Test T2 | |
|---|---|---|---|---|---|---|
| | Img | seq | img | seq | img | seq |
| 2CH | 318845 | 1033 | 600 | 120 | 20 | 4 |
| 4CH | 706160 | 1433 | 880 | 176 | 95 | 19 |
| A-5CH | 166955 | 354 | 205 | 41 | 15 | 3 |
| A-RV | 196266 | 347 | 450 | 92 | 10 | 2 |
| APLAX | 397240 | 775 | 535 | 107 | 79 | 14 |
| NO-ORGAN | 84815 | 305 | 380 | 76 | 0 | 0 |
| P-SAX | 288171 | 904 | 800 | 160 | 0 | 0 |
| PLAX | 383551 | 844 | 915 | 183 | 105 | 21 |
| SAX-AV | 222246 | 828 | 350 | 70 | 0 | 0 |
| SUBCOST. | 107325 | 171 | 5 | 1 | 5 | 1 |
| SUM | 2871574 | 6994 | 5130 | 1026 | 320 | 64 |

## 4.3 Sampling of training subsets

The experiments have been carried out using two different strategies for sampling from the training data; *image-based* and *sequence-based* sampling.

Using the image-based scheme, we sample randomly from all images, including augmented versions. This strategy allows for the use of a large and diverse selection from the training set.

When working with cardiac ultrasounds, it is however more realistic to sample entire sequences rather than individual images, since a full sequence is al-

ways available. We have therefore also performed experiments where we sample entire sequences (including augmentations) rather than just single frames. The sequence-based sampling has, differently from the image-based sampling, been performed in a balanced way, sampling the same number of sequences from all classes. As some classes are much smaller, this means the total number of samples in the largest subset sampled from sequences is much lower than for the largest subset sampled from random images.

## 4.4 Implementation details

The models all take three channels (RGB) as input, and the grayscale ultrasound images are therefore represented by three (equal) channels. The images are then center cropped from 256x256 to 224x224.

All models except the specialized ResNet model were fetched from Huggingface using the initial weights that were included there.

The models were all finetuned and trained on a single GPU, an NVIDIA RTX 4000 Ada, where the number of epochs was set to ensure that the model processed at least $2^{14}$ (16384) samples in total, independent of the size of the training set (for more details see the Appendix).

## 4.5 Evaluation metrics

We used total classification accuracy as the evaluation metric for prediction performance in all experiments. In view recognition, no single error type is more costly than another (unlike, for instance, in diagnostic classification); thus, the main objective is to minimize the total number of errors and, consequently, the number of corrections the clinician needs to do. Since the distribution of samples in our dataset reflects the typical clinical situation – where some views are collected more frequently than others – we found total accuracy to be a well-suited comparison metric. The metric then directly reflects the model's performance on the expected, real-world data distribution.

## 4.6 Experiments with linear probing

In these experiments we have tested the performance of the specialized ResNet model compared to that of the five different foundation models when training on increasing amounts of labelled data from our downstream task of view recognition. We kept the foundation models and their resulting embeddings unchanged; only the Support Vector Machine (SVM) classifier that was applied on top of them was trained with the labeled data.

The experiments have been carried out using two different strategies for sampling from the training data; *image-based* and *sequence-based* sampling. For the experiments on linear probing with sequence-based sampling we created multiple datasets with the same size, but with different random seeds, to get an impression of the variation due to sampling differences. For the other more computationally heavy experiments, only one seed point was used creating one sampled training set.

The results from the experiments are shown in Figure 1 and Figure 2. The X-axes, showing the amount of labelled training data, are for all plots shown on a logarithmic scale to better visualize the performance with low numbers of training samples.

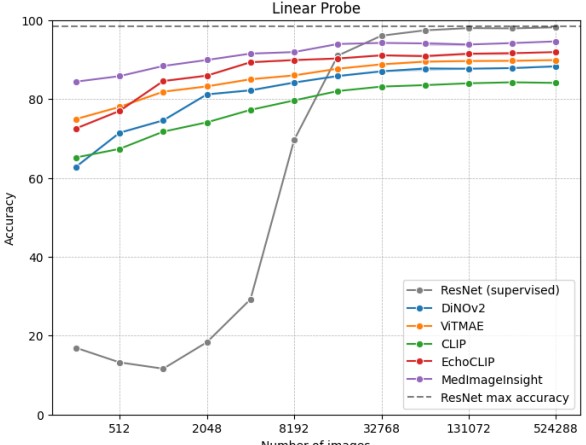

**Figure 1.** Results of linear probing with image-based random sampling. (Note: logscale on x-axis).

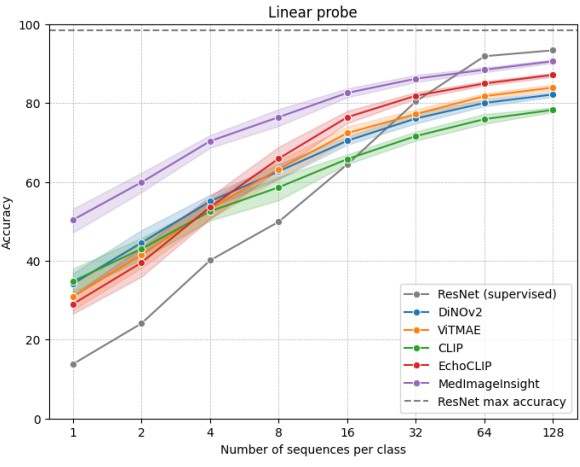

**Figure 2.** Results of linear probing with sequence-based balanced sampling. The shaded area represents the 95% confidence interval for the mean at each x-value, calculated from 20 different random seeds (Note: logscale on x-axis)

What we see from these experiments is that using frozen foundation models with a simple SVM classifier can achieve a reasonable performance even with minimal labeled training data. In these scenarios with limited labelled training sets, the foundation

models significantly outperform a specialized ResNet model. However, while the foundation models outperform the ResNet model with limited data, the ResNet model closes the performance gap and surpasses them as the labeled training set becomes larger. We see that the more specialized models, MedImageInsight and EchoCLIP, perform better than the general models. Notably, MedImageInsight performs slightly better than EchoCLIP in the linear probing scheme. This is likely because MedImageInsight is a larger model with a larger embedding dimension, allowing it to encode more diversity.

For the sequence-based sampling the X-axis gives the number of sequences that were sampled per class. All frames for each sequence, including augmentations were then used, which approximates around 300 images per sequence. Hence, the X-axis here corresponds to a range of about 3000 to 400.000 images. Comparing the results from the sequence-based sampling with that of the image-based, we see that the performance relative to the number of images is much higher for the image-based sampling. This shows that the models benefit more from a diverse training set than a large training set.

## 4.7 Experiments with full finetuning

The same experiments as for linear probing were carried out with models where full finetuning was used instead, but comparing the same selection of models, the same sampling schemes and the same amounts of training data. The results are shown in Figure 3 and Figure 4 for image-based and sequence-based sampling respectively. Again, the amounts of training data are shown on a logarithmic scale.

We see from these experiments that with this full finetuning of the foundation models, high performance is reached already for very low amounts of labelled training data. Again, we see that the more specialized foundation models perform slightly better than the general models. In contrast to what we observed for the linear probing, we observe that the foundation model specialized on cardiac ultrasound, EchoCLIP, here perform slightly better than the large medical model, MedImageInsight.

Using finetuning rather than linear probing is somewhat more demanding in terms of both the work and training time required. The reward is that higher performance is achieved for lower amounts of labelled training data.

## 4.8 Comparing maximum performance

In the following experiments we have compared the maximum performance of a selected group of models after they were supervisedly fine-tuned on the entire training set of approximately 2.9 million sam-

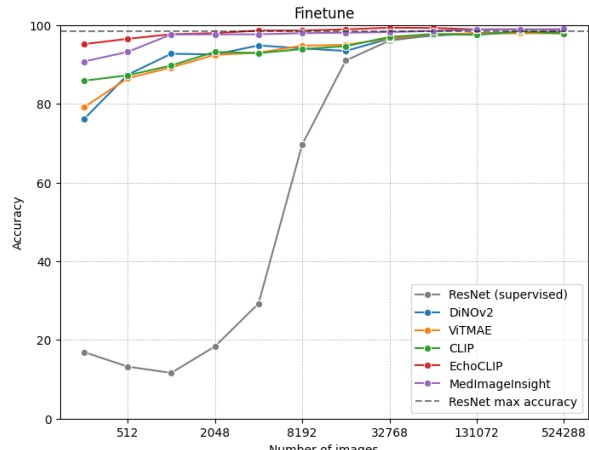

**Figure 3.** Results of full finetuning with image based random sampling. (Note: logscale on x-axis)

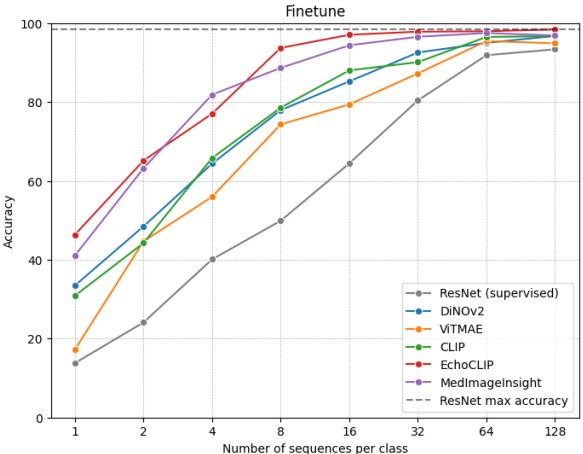

**Figure 4.** Results of full finetuning with sequence-based balanced sampling. (Note: logscale on x-axis)

ples (including augmentations). We performed these comparisons on both the large test set (T1) used in the previous experiments and the smaller, more challenging test set (T2).

We selected four models, each representing a different level of specialization for cardiac ultrasound view recognition: DINOv2 as a general model, MedImageInsight as a general medical model, EchoCLIP as a cardiac ultrasound model, and the ResNet50 model specifically trained for this task. Table 3 presents the results of this comparison, reporting accuracies at both the image and sequence levels. For sequence-level accuracy, we used a majority vote over the five frames that represent each sequence.

From the T1 results we see that all models achieved very high accuracy, with the foundation models performing marginally better than the task-specific model. The performance of the foundation models also slightly increased with greater specialization toward the specific domain. The results for the set of difficult cases (T2) are given in the two

rightmost columns of Table 3, and here we see a quite different situation. For these data the most specialized foundation model, EchoCLIP, actually has the lowest performance of all the foundation models, and performs even slightly worse than the task-specific model when evaluated at sequence level. Furthermore, the large MedImageInsight model for medical images, shows a significantly lower performance than the general DINOv2 model.

For the T2 test set, we also observe a significant increase in performance when using all five images from a sequence compared to single images. This is a reasonable finding, as these cases represent people with very specific conditions where a sequence can provide valuable contextual information that a single image may not convey.

**Table 3.** Results for models trained on the entire dataset, for both our testset of typical cases (T1) and our testset of difficult cases (T2).

| Models | T1 accuracy | | T2 accuracy | |
| --- | --- | --- | --- | --- |
| | img | seq | img | seq |
| DINOv2 | 98.4% | 99.1% | 85.6% | 92.2% |
| MedImageInsight | 99.1% | 99.7% | 78.1% | 84.4% |
| EchoCLIP | 99.3% | 99.9% | 71.3% | 75.0% |
| ResNet50 | 98.4% | 98.9% | 70.6% | 76.6% |

Choosing the right model may not always be only about performance in terms of accuracy. We have therefore also compared the characteristics of these models in terms of both size and computational requirements in relation to accuracy. Figure 5 illustrates this for both T1 (typical cases) and T2 (abnormal cases). The ResNet model is the smallest and fastest and for the typical cases (T1) the performance is also on level with the larger models. For the abnormal out-of-distribution cases we see that the DINOv2 model gives the best accuracy as well as providing the best trade-off when taking speed and size into account.

## 5 Discussion

A major finding through our experiments is that foundation models significantly outperform the specialized ResNet model when trained with very limited labeled data. This demonstrates the power of transfer learning from large-scale pretraining, which provides a strong starting point. Even with a simple linear classifier (SVM) and frozen foundation model, the models achieve reasonable performance with very few samples, highlighting that the learned features from pretraining can be highly effective.

For linear probing, the more specialized models perform better than the general ones. However, the larger MedImageInsight model performs slightly

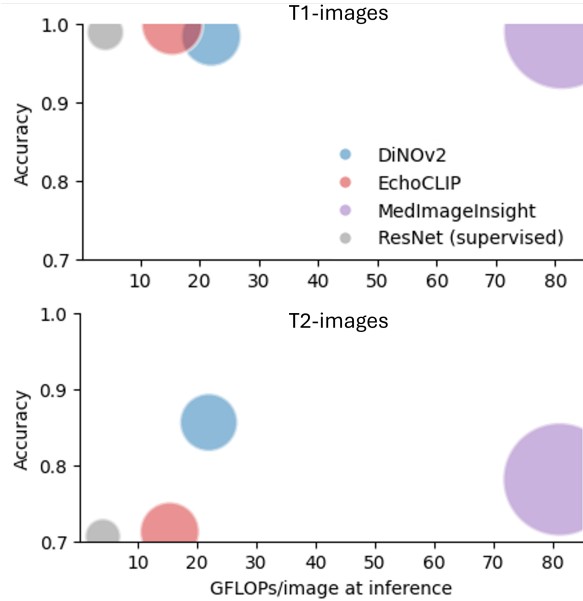

**Figure 5.** Inference time and model size (in terms of the number of parameters) vs accuracy for experiments on the two test sets with typical images (T1, top) and abnormal (T2, bottom).

better than EchoCLIP. This is likely due to its larger size and embedding dimension, which enables it to capture more diverse features.

This trend shifts with full fine-tuning where the most specialized model, EchoCLIP, which is specifically trained on cardiac ultrasound, is slightly better than the general medical model, MedImageInsight. This suggests that when models are allowed to update all their parameters, the domain-specific pretraining becomes a more significant advantage. In general, we see that the full finetuning, as expected, yields higher performance than linear probing, particularly with lower amounts of labeled data. The trade-off here is that the full fine-tuning is more time-consuming and computationally demanding.

The results on the T1 test set with models fully finetuned on the entire training set, show that regardless of the model choice, if sufficient training data is available, all the models including the task specific ResNet can reach a very high accuracy of near or above 99%. The most domain-specific foundation model, EchoCLIP, gives a marginally higher accuracy than the others.

In contrast, on the T2 test set with unusual, difficult cases not well represented in the training dataset, we see a more unexpected result where the general foundation model, DINOv2, performs significantly better than the domain-specific models EchoCLIP and MedImageInsight. This is an interesting finding, which suggests that models pretrained on a broad range of natural images may have a better ability to generalize and handle ambiguous or difficult-to-classify cases. This could be because the

diversity of features learned from a massive, varied dataset helps prevent overfitting.

This result suggests that selecting a domain-specific foundation model is not always beneficial. When presented with atypical or out-of-distribution medical images, generalist models such as DINOv2 may be more robust. This also shows the importance of using diverse evaluation sets, where systematic evaluation also on challenging, out-of-distribution test sets can be key to assessing the models' reliability in real-world clinical scenarios. It should be noted that in our experiments further optimizing the hyperparameters when finetuning the foundation models or changing the dataset sampling could influence the accuracy levels. Hence, the reported results should be viewed as overall trends.

## 6  Conclusion

We have compared the performance of several foundation models against a specialized ResNet model. Our results show that foundation models achieve superior performance when limited amounts of labeled training data are available, a common challenge in medical imaging.

The highest performance for the foundation models is achieved when they are fully finetuned to the downstream task, but this comes at a computational cost. Still, we have shown that these models can be fine-tuned and run on relatively limited GPU hardware. Hence, high performance in specialized tasks is achievable without the need for extensive computational resources.

Generally, the more domain-specific foundation models have shown better performance than the general models in our experiments. However, we observed an important exception to this for more difficult out-of-distribution cases, where the more general model had a greater ability to generalize and perform well.

In conclusion the choice of a foundation model should be guided by the specific application's needs. For scenarios with limited labeled data and a focus on standard, in-distribution cases, a fine-tuned domain-specific model can be the best choice. Still, if enough labelled training data are available a task-specific model competes well in terms of accuracy, while being both faster and smaller. However, for applications requiring a model to contend with a variety of difficult or out-of-distribution cases, a generalist foundation model may be more reliable.

## Acknowledgements

This study was partly funded by project grants from the Research Council of Norway (#309439 #313756).

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

# Appendix

This section gives some more details on the models and the parameters used for training.

## Linear Probing

For linear probing, we used the default parameters for sci-kit learn's Linear Support Vector Machine functions (LinearSVC): L2 norm for penalization and squared Hinge loss.

## Full Finetuning

All model weights, except the specialized ResNet model, were downloaded from HuggingFace (https://huggingface.co/). We used largely the default parameters in the "Trainer" class from Hug-

gingFace's transformers library to run the training for all the foundation models, including the default optimizer AdamW. The learning rates for each model are included in Table 4.

**Table 4.** Overview of models, source (under https://huggingface.co/) and learning rates

| Model | Source | L-rate |
|---|---|---|
| DINOv2 | facebook/dinov2-base-imagenet1k-1-layer | $5.0 \cdot 10^{-6}$ |
| ViTMAE | facebook/vit-mae-base | $2.0 \cdot 10^{-4}$ |
| CLIP | openai/clip-vit-base-patch16 | $1.0 \cdot 10^{-5}$ |
| EchoClip | mkaichristensen/echo-clip | $5.0 \cdot 10^{-5}$ |
| MedIm.Ins. | lion-ai/MedImageInsights | $5.0 \cdot 10^{-6}$ |

No pre-processing or specific adaptations to echocardiography was performed apart from the center-cropping, repeating the single channel if a model demanded a three-channel input and using the same normalization as in the original model pre-training. ViTMAE, DINOv2 and MedImageInsights normalize the input using ImageNet's means and standard deviations. CLIP and EchoCLIP normalize the input using WebImageText's means and standard deviations. For all models these are the means and standard deviations that accompany the models upon download from HuggingFace. Augmentations consisted of slight angle tilts and resampling of widths.

We adjusted the number of epochs based on the dataset size to ensure that the models processed at least $2^{14}$ (16384) samples. Table 5. shows the number of epochs used for each dataset derived using the image-based sampling scheme, and table 6. shows the number of epochs used for the datasets derived using the sequence-based sampling scheme.

**Table 5.** Overview of the number of epochs used when finetuning on the datasets derived using the image-based sampling scheme, where the number of epochs is reduced with higher numbers of images. For 16384 images and upwards only one epoch is used.

| #images | 256 | 512 | 1024 | 2048 | 4096 | 8192 | 16384 → |
|---|---|---|---|---|---|---|---|
| #epochs | 64 | 32 | 16 | 8 | 4 | 2 | 1 |

**Table 6.** Overview of the number of epochs used when finetuning on the datasets derived using the sequence-based sampling scheme, along with the number of sequences and the number of images in the datasets.

| #seq per class | 1 | 2 | 4 | 8 | 16 | 32 | 64 | 128 |
|---|---|---|---|---|---|---|---|---|
| #epochs | | 5 | 3 | 2 | 1 | 1 | 1 | 1 |

