# OpenReview forum: "Comparing Foundation Models for Medical Images: A Study on Limited Data and Generalization"
_NLDL.org/2026/Conference — NLDL 2026 Poster_

### Official Review · Reviewer_p5JL · 2025-09-17
**Important field but limited novelty**

**Rating:** 2
**Confidence:** 5
**Final Rating:** 4
**Final Confidence:** 5

**Summary:**

This paper investigates how vision foundation models pretrained on different domains compare with a task-specific supervised ResNet model for cardiac ultrasound view recognition. The study examines two dimensions: (i) performance as a function of labeled training set size, and (ii) generalization to out-of-distribution or difficult cases. The authors evaluate several well-known general-purpose models (ViTMAE, DINOv2, CLIP) and two medical-oriented models (MedImageInsight, EchoCLIP) under both linear probing and full fine-tuning. Results show that foundation models outperform the task-specific model when labeled data are scarce, domain-specific models perform best in limited-data settings, while general models generalize better to unusual cases. With sufficient labeled data, however, the ResNet achieves near-parity in accuracy, while being smaller and faster.

**Strengths:**

- Addresses an important and timely problem: medical image classification under limited labeled data.
- Provides a clear experimental comparison across multiple foundation models and a supervised baseline.
- Considers both linear probing and full fine-tuning strategies.
- Practical insight: highlights trade-offs between model specialization, data availability, accuracy, and computational cost.
- The finding that general models may generalize better on atypical cases is potentially interesting.

**Weaknesses:**

- The main findings (foundation models excel with scarce labels, fine-tuning boosts performance, supervised models can match with sufficient data) largely confirm what is already well established in prior work.
- Recent and highly relevant ultrasound-specific foundation models such as USFM, UltraSam, and FetalCLIP are not cited or compared, despite their strong relevance to the task. Their omission weakens the contribution.
- Much of the methods section is devoted to summarizing well-known architectures (ViTMAE, CLIP, DINOv2, etc.) rather than describing adaptations specific to echocardiography. Details on preprocessing, augmentation, and experimental controls are thin.
- Definition of “difficult cases” is vague; the OOD test set (T2) is not clearly characterized, making generalization claims less convincing.
- The authors acknowledge that finetuning results could vary with different hyperparameters or sampling strategies, but do not explore this systematically.
- While ResNet is described as smaller/faster, no quantitative runtime or memory comparisons are provided, which limits the practical value of efficiency claims.
- Discussion focuses almost exclusively on classification accuracy, without considering calibration, robustness, or clinical utility — critical aspects in medical imaging.

**Final Justification:**

The rebuttal effectively clarifies several previously unclear aspects, including dataset characterization, preprocessing, and model selection rationale. While the paper does not introduce methodological novelty, it provides a systematic and well-controlled empirical comparison of foundation models for cardiac ultrasound view recognition, an area of growing practical interest. The authors’ discussion of the T2 out-of-distribution results and their transparent handling of limitations strengthen the work’s value as a useful reference study for researchers dealing with limited labeled data in medical imaging.

Overall, this is a carefully executed and informative study whose contribution lies in empirical clarity rather than innovation. It is suitable for acceptance as a solid, practical paper of community interest.

**Justification:**

The paper is well executed and provides a clear side-by-side comparison of foundation models versus a supervised baseline in a focused medical imaging task. The experiments confirm known trends and may be of practical interest to those working with limited labeled data in ultrasound. However, the novelty is limited, the related work omits very relevant ultrasound-specific foundation models, and the methodological depth is modest. The discussion highlights descriptive trends but lacks deeper analysis of efficiency, robustness, or clinical implications. Overall, the work is a useful case study but falls short of originality and comprehensiveness required for a stronger accept.

---

> ### Author Rebuttal · Authors · 2025-10-22
>
> We thank the reviewer for the detailed remarks and for pointing out the strengths of the paper highlighting the challenge of medical image classification under limited labeled data by providing a clear experimental comparison of multiple foundation models and a supervised baseline, using both linear probing and full fine-tuning. And also for further pointing out that the study offers practical insights into trade-offs between specialization, data, accuracy, and cost.
>
> Below we would like to comment on the remarks the reviewer made on the weaknesses.
>
> **Regarding novelty**: We agree with the reviewer that many of the assumptions made in this study are already established for natural image and other medical imaging tasks, and that there is no methodological/architectural novelty. However, all aspects of the use of FMs are not systematically covered and for our case of cardiac ultrasound, we wanted to dive into some specific aspects of interest. We believe that comparing different FMs applied to cardiac ultrasound with well-defined experimental settings provides practical insights to the community, especially in the light of limited data and out-of-distribution evaluation. While our results on the T1 test dataset are consistent with the common assumption that domain-specific FMs will outperform general FMs, our results on the more difficult T2 dataset challenge this expectation. We believe this result is particularly interesting to the community, as medical models are most commonly tested on datasets which may be from different institutions but otherwise resemble the training data.
>
> **Regarding suggested ultrasound models**: We thank the reviewer for mentioning several relevant models that we did not include in the relevant work section. We will do so in the camera ready version. Due to practical constraints, we limited our model selection to a diverse set of models that were open source and accessible on Huggingface in the following distribution categories: general natural images, general medical images and echo images. Of course, this limits the selection, and we acknowledge that the reviewer has listed some recent impactful and relevant models. USFM is a strong, well-performing FM for general ultrasound which includes some undisclosed echo data of unknown source. Since we could not guarantee whether it already contains in-domain data and wanted to have well defined experimental settings, and since it is not available on HuggingFace, we decided that it was out-of-scope for this study and left it for future work. As for the other two models listed, FetalCLIP is indeed available on Huggingface, but it focuses on fetal ultrasound images, which are not present in our dataset. UltraSAM is designed for segmentation, and we therefore considered it less relevant for our classification task. However, we agree that it would make sense to explore additional in-domain or near domain medical FMs (such as the ones the reviewer listed) in the future.
>
> **Regarding pre-processing**: In our experiments, no pre-processing or specific adaptations to echocardiography was performed apart from the center-cropping mentioned in the paper, repeating the single channel if a model demanded a three-channel input and using the same normalization as in the original model pre-training. Augmentations consisted of slight angle tilts and resampling of widths. We are happy to revise the manuscript to make these details more clear.
>
> **Regarding hyperparameter tuning**: The reviewer is correct in pointing out that we did not explore hyperparameter tuning or sampling strategies systematically. A main reason for this was that we wanted to see how these methods can be made to work more or less out-of-the-box, and look into their relative performance rather than their optimal one. We also considered systematic evaluation of multiple settings for all methods and sizes of training sets to reach individual optimal performance to be very difficult within reasonable time.
>
> **T2 dataset characteristics**: We agree with the reviewer that the difficult test set T2 was not clearly characterized. Difficult cases include abnormal anatomies, pathologies, and low image qualities. We will revise it in the camera-ready version.
>
> **Regarding runtime and memory comparisons**, we chose to compare the models in terms of GFLOPS and number of parameters, as specific runtime and memory can be GPU and platform dependent. However, we agree with the reviewer that measurements such as memory footprint and speed can be valuable to readers. They can be included in a revised version of the manuscript.
>
> **Regarding classification accuracy as a metric**, we chose to focus on this as our main objective was to investigate how performance will vary with the size of the training set for different types of models. In the case of view recognition, no errors are more costly than others, so we believe classification accuracy serves well as a metric for comparison when our class distribution represents the distribution typically met in a clinical situation. We agree with the reviewer that robustness testing is critical in the medical domain and could have been emphasized more in our manuscript. Our tests on the out-of-distribution T2 test dataset provide some insight into the models’ robustness, but this dataset is limited, and further robustness testing is an important direction for future investigations. We also acknowledge the importance of calibration for clinical deployment. However, since our work was not centered on optimization or deployment readiness, we consider calibration analysis to be beyond the intended scope of this study.

---

### Official Review · Reviewer_tvQC · 2025-10-07
**A nice experimental study with an interesting finding**

**Rating:** 4
**Confidence:** 4
**Final Rating:** 4
**Final Confidence:** 4

**Summary:**

The paper investigates the performance of fine-tuned foundation models compared to a specialized model. Performance is evaluated on the task of viewpoint identification from cardiac ultrasound images (classification with 10 classes). The foundation models chosen are all trained with self-supervised learning (SSL) and include both generalist models (ViTMAE, DINOv2 and CLIP) as well as a medically focused (MedImageInsight) and domain adjacent model (EchoCLIP, pre-trained on cardiac ultrasound data). The specialized model is a ResNet-50.

Experiments are done with linear probing (training a linear SVM on the extracted features) and full finetuning. Two test sets are used: T1 using in-distribution “normal” images and T2 using out-of-distribution images containing difficult abnormal pathologies. Results on the T1 test data show that for low amounts of training data, the foundation models outperform the ResNet-50. For large amounts of data, the ResNet-50 outperforms the foundation models for linear probing and achieves comparable performance for full fine-tuning. For the foundation models, full finetuning achieves much higher accuracy even for low amounts of training data. Finally, when evaluated on the T2 dataset, the generalist DINOv2 model outperforms the medically focused models, which is an interesting surprise.

**Strengths:**

1. The paper is well written. The text is clear and steps are well motivated.

2. It is great that performance is evaluated on *both* in-distribution and out-of-distribution tasks. It is also good that test sets are from separate sites compared to the training data.

3. That full fine-tuning performs as well as it does and that DINOv2 can outperform domain specific foundation models for difficult cases are nice findings worth sharing.

**Weaknesses:**

1. It would be great to show some example images of the data. Both for the “normal” cases and the more difficult “T2” cases. This would help a reader assess if their data is similar and therefore they may expect similar results.

2. L060-062: That these findings will transfer to other tasks and domains is, to me, a strong statement. It would be great if the authors could further motivate this or rewrite to a less strong statement.

3. L112-118: Nitpick: Must foundation models be trained using SSL? Is, e.g., SAM not a foundation model? Some readers may have different notions on what foundation models are. It would be great to clarify that this paper focuses on SSL models (which is completely good and fair) in the abstract.

4. Sec. 3.2.1: Question: Is the ResNet50 pre-trained on, e.g., ImageNet or is the training completely from scratch with random initial weights?

5. Sec. 3.4.2: How were the models specifically finetuned? Please provide a full description of the training: learning rate, number of epochs, optimizer, etc. This may be in an appendix to save space.

    * Were the models trained to convergence or with a “fixed” compute budget?
    * I’m only asking because the specialized models may have an advantage in that they are already “most of the way” there compared to DINOv2, so will reach their highest accuracy with fewer epochs. Therefore, *if* the compute budget is small, maybe we are not seeing the full potential of DINOv2?
    * Note: a limited compute budget is also valid, I am *not* asking to re-do the experiments. This is simply useful info for a reader in order to asses whether the results in the paper are applicable to their case

5. Sec 4.3: Why was class-balanced sampling not performed for image-based sampling? I am asking specifically because I find it curious that the ResNet performance *decreases* in the beginning as more images are added. Could this be linked with high class imbalance for the small datasets?

6. Secs 4.6 and 4.7: In line with the previous comment, it would be interesting to know the errors are distributed among the different classes. Ideally, with confusion matrices (could be in an appendix) but a text comment can suffice. Is it the same for all models? Does it match the class imbalance? Note: I realize this may require extra experiments, so it is not crucial to me that the authors answer this fully. I am just curious!

7. L390: Minor: For reproducibility, please provide a citation. Ideally, list the specific name/link of each downloaded model. This may be an appendix.

8. L399: What is “total accuracy”? Is it just “normal unweighted accuracy”?

9. Figs. 2 and 4: What does the “ResNet max accuracy” line signify? Is it the max performance from the image based sampling? Or was the sequence based sampling continued to larger sizes for the ResNet?

10. How much time was spent on full finetuning vs linear probing? Would be interesting to know to more fully understand the trade-off since full finetuning performs so much better.

**Final Justification:**

I thank the authors for addressing some of my concerns - especially the statement regarding the transferability of the results.

I remain enthusiastic about acceptance since I think the results are worth sharing at NLDL.

**Justification:**

The paper is well written and presents a nice systematic study with useful and a, to me, surprising finding. I have only minor weaknesses and questions - **none** of which are deal-breakers. My most important concerns to address are those related to reproducibility, as this is a very experiment focused study.

---

> ### Author Rebuttal · Authors · 2025-10-22
>
> We thank the reviewer for their valuable input, addressing the strengths and weaknesses of our paper.
>
> **Example images**: We agree with the reviewer that example images showing the classes and variations would be helpful for the reader. Unfortunately, for privacy reasons, we are unable to include such examples. We would however like to add that the difficult T2 test dataset has large pathological variations, and one example image would be just that and not give a general picture of the variations.
>
> We also agree that our introduction makes a **strong statement regarding transferability** to other tasks or domains that we have not substantiated properly. We will therefore rewrite this part with a less strong statement.
>
> **Model initialization and hyperparameters**: We are happy to add further information regarding model initialization (ResNet was indeed initialized with ImageNet weights), learning rates, number of epochs and optimizer, as well as links to each downloaded model. We selected the number of epochs to ensure that the model processed at least 2^14 samples, thus the number of epochs varied depending on the size of the training dataset. The models were trained with the same number of epochs if the size of the training dataset was the same. For example, all models were trained for 64 epochs on the dataset with 256 images, and for 32 epochs on the dataset with 512 images.
>
> To answer some of the reviewer’s questions:
>
> **Regarding class-balanced sampling**: our training and test datasets have a class imbalance which reflects the real world scenario in a clinic where data from some views are always collected more frequently than others. Our image-based sampling case will thus reflect the true data distribution, while sequence-based sampling is class-balanced. However, the reviewer raises a valid point that for small datasets, poor performance (such as the ResNet performance decrease) can be linked with the high class imbalance.
>
> Yes, **"Total accuracy"** is the total number of correctly classified samples divided by the total number of samples. It is not weighted.
>
> The **ResNet max accuracy** line signifies the performance for the ResNet model when trained on all training samples.

---

### Official Review · Reviewer_y93V · 2025-10-07
**Very well written paper, but limited novelty**

**Rating:** 2
**Confidence:** 4
**Final Rating:** 4
**Final Confidence:** 4

**Summary:**

The paper investigates the performance of five foundation models and compares them to a specialized Resnet model (i.e the specialized model is not pretrained). The application is cardiac ultrasound view recognition.
The models are tested without finetuning, using only a linear probe, and with full finetuning. The authors find that foundation models achieve superior performance when limited amounts of labeled training data are available. They also find that the more domain-specific models have better performance compared to the general models. The Resnet model is generally outperformed in all cases compared to the foundation models.

**Strengths:**

The paper addresses the challenge in medical imaging with limited labeled data, which is of general interest.

It includes a comprehensive set of foundation models (ViTMAE, DINOv2, CLIP, MedImageInsight, EchoCLIP), and tests the two common cases of linear problem and full finetuning.

The primary dataset comes from four different hospitals, where the data from three of the hospitals were used for training, and testing was performed on the fourth hospital. This makes the study much more interesting as we then have a realistic generalization performance assessment.

The paper is very well written and easy to follow.

**Weaknesses:**

There are some issues with the paper, but my main concern is actually novelty. The dataset is very nice, but undisclosed. The hyperparameters for the different models are undisclosed, and the code is not available.

This is coupled with the main finding, which is that pretrained foundation models perform much better than models that are only trained on limited data. This conclusion is expected, and it is well established that pretrained models outperform models that are not pretrained. It is also well established that foundation models work quite well coupled with a linear probe.

Therefore, the new insights from this paper are quite limited.

A few other remarks:
Figure 2 has 95% confidence intervals; however, they are not plotted in Figure 1. Why?

It would be helpful if we could understand how many images make up "one sequence per class" so that Figures 1 and 2 are directly comparable for the reader. I think one sequence means five images (indicated under 4.2 dataset), but I'm actually not sure.

Could Table 3 have confidence intervals, so that the models can be compared using statistical testing?

**Final Justification:**

The authors sufficiently argued their case in the rebuttal.

**Justification:**

My main reason for rejecting this paper is the lack of novelty. Data and code are not available online, and the scientific findings are well established already. The main merit that models are compared on a closed, realistic dataset, but given the lack of transparency, I only see limited value in this paper.

If the authors could share code and data, it would be of much higher value.

---

> ### Author Rebuttal · Authors · 2025-10-22
>
> We thank the reviewer for pointing out the value of the task and the selection of the dataset splits and also for raising constructive feedback. Below we answered the remarks that were raised:
>
> **Open code and datasets**: We highly appreciate the reviewer pointing out the importance of open code and open data. One goal of this study was to explore the use of easily accessible and openly available foundation models hence we picked Hugging face as the base of the experiments. Unfortunately, in the medical domain, open data is still an issue in some areas, with cardiac ultrasound  being one of them. There is no established and suitable open dataset for the view recognition task and authors from the popular FMs covering that particular in-domain data do not disclose their data. Due to patient privacy regulations, we are currently not allowed to publish our internal datasets. We also believe that our inclusion of a more difficult test dataset provides added value. Such datasets are rarely included to evaluate models in the medical domain, where test sets typically differ by institution but otherwise resemble the training dataset. Our result on this dataset challenges the common assumption that domain-specific FMs will outperform general FMs. While this assumption holds for our T1 test set, which resembles the training dataset, a general FM significantly outperforms the specialized models on the atypical test data.
>
> **Hyperparameters and code**: We agree with the reviewer that information about hyperparameters is lacking in our paper. Should the paper be accepted, we will ensure that hyperparameters for the different models are specified in the camera-ready version. As for the code, it consists of simple scripts, building on HuggingFace’s transformers library, for finetuning and inference, which can be shared. The specialized part of the code is related to the dataloader and samplers for images and sequences, and has limited value without the data.
>
> **Regarding the number of images per sequence**: there are always 5 images per sequence for the test set, where these have been specifically sampled to represent one heart cycle with a limited number of frames. For the training set, each sequence covers at least one full heart cycle with the original full time-resolution and the number of frames per sequence is therefore both much higher and varies considerably between patients. On average, there are around 100 frames per sequence. In addition, comes four augmented versions of each image in the full training dataset (2 images augmented with tilt and 2 images augmented with width variations). We agree that this was unclear in the manuscript, and we will revise the text accordingly.
>
> **Regarding confidence intervals in Figure 2**, these were computed by training multiple instances of the models with different seeds for dataset subsampling. The seed and thereby the choice of samples, can have a large effect when limiting the sampling to include images only from very few sequences (down to 1 in Figure 2), but less effect when the same number of images are sampled randomly across sequences. We therefore wanted to look into this specifically for the sequence-based sampling, while we considered it less important for the image-based sampling.
>
> We agree with the reviewer that **adding confidence intervals for Table 3** would be useful for statistical testing. Should the paper be accepted, we will update this table in the revised version of the manuscript to include confidence intervals.

---

### Official Review · Reviewer_mVyE · 2025-10-08
**A Thorough Empirical Comparison of Foundation Models for Medical Imaging with insights into OOD generalization**

**Rating:** 4
**Confidence:** 4
**Final Rating:** 4
**Final Confidence:** 4

**Summary:**

The paper focuses on the empirical study comparing the performance of various vision foundation models in the low-data context of medical imaging. The selection of vision foundation models ranges from general-purpose (DINOv2) to domain-specific (EchoCLIP) against a task-specific supervised baseline (ResNet-50). Furthermore, the study takes on two main directions: 1. how performance varies as a function of the amount of labeled training data, and 2. how these models generalize to difficult, out-of-distribution (OOD) examples.

While the comparison of foundation models on downstream tasks is an established area of research, the paper's primary contribution is not the introduction of a new method but rather its systematic and comprehensive empirical analysis. The novelty lies in the direct comparison across a spectrum of models, from general to highly specialized, and most importantly, the inclusion of a challenging out-of-distribution test set (T2).

The study is validated using experiments on a 10-class cardiac ultrasound view recognition task. It uses T1 for typical in-domain cases and T2 for difficult OOD cases from a separate clinical site. They also assess the performance using both computationally efficient linear probing and comprehensive full fine-tuning.

The results are properly presented and backed by results. They show:
 1. Foundation models significantly outperform the task-specific ResNet in low-data regimes.
2. The ResNet closes the performance gap and becomes competitive when sufficient labeled data is available.
3. The DINOv2 model exhibits superior generalization on the difficult T2 test set, significantly outperforming the highly specialized EchoCLIP and even the task-specific ResNet.

**Strengths:**

1. Rigorous methodology and Quality:

The experimental design is comprehensive and robust. The selection of foundation models is justified, ranging from general to highly specific models. They evaluate two distinct adaptation strategies (linear probing and full fine-tuning) while analyzing performance across logarithmically scaled subsets of the ~2.9 million image training set. The inclusion of two distinct test sets (T1 and T2) is a major strength. This provides deeper insight into the practical reliability of different models.

2. Significance of result:

The paper's most significant contribution is the counterintuitive finding that the generalist DINOv2 model outperforms domain-specific models on the challenging T2 test set. This challenges the prevailing assumption that maximum domain specialization is always optimal for medical imaging tasks. This could provide valuable guidance for developing models robust to real-world clinical data.

Q: Could you elaborate on your hypothesis for DINOv2's superior performance on the T2 dataset beyond the diversity of pre-training data (like architectural choices)?

**Weaknesses:**

1. Limited Conceptual Novelty

The general approach of comparing foundation models on downstream tasks is well-established. The contribution is therefore more empirical and confirmatory rather than the introduction of a new method, algorithm, or theoretical framework.

2. Limited Scope of evaluation:

All experiments are conducted on a single classification task (view recognition) and a single modality (cardiac ultrasound). The paper asserts that its "findings should be transferable," but this claim is not empirically validated within the study. The generalization of these findings to other tasks (e.g., segmentation) or modalities (e.g., CT, MRI) remains an open question.

3. Potential for Stronger Evaluation Metrics:

The study relies on total accuracy as its performance metric. Given the notable class imbalance (4CH vs NO-ORGAN in Table 2), relying solely on total accuracy can be misleading. Incorporating metrics such as balanced accuracy or class-wise F1-scores would provide a more complete understanding of model performance, particularly for under-represented classes.

**Final Justification:**

The paper is limited by its conceptual novelty, where it does not establish a new framework or a new state-of-the-art, while exhibiting some limitations regarding the scope of its evaluation. Considering its clear contribution in terms of the interesting finding backed by methodical experiments and clear presentation, I lean towards acceptance.

**Justification:**

The paper presents a methodologically sound empirical study that delivers a highly significant and practical finding regarding the out-of-distribution generalization of foundation models in medical imaging. Since the result challenges conventional belief, it outweighs its limited conceptual novelty, making it a potentially valuable addition to the conference proceedings.

I would recommend that the authors expand the scope of evaluation to multiple tasks and/or datasets to strengthen their claims.

---

> ### Author Rebuttal · Authors · 2025-10-22
>
> We thank the reviewer for the thoughtful review and for highlighting the strengths of the study. Below we tried to answer the posed questions and comment on the raised weaknesses:
>
> **Weaknesses remark 1 and 2**: We agree that our work is primarily empirical rather than introducing a novel methodological framework. However, we believe such empirical investigations remain valuable to the community. While the general idea of comparing foundation model performance is indeed well established, all aspects are not necessarily systematically covered. For our case of cardiac ultrasound we wanted to dive into some specific aspects of interest. While we do not demonstrate transferability to other tasks or modalities, our study establishes a foundation and reference point for future investigations because it selected models that are easily accessible for reproduction. Furthermore, as the reviewer points out, our results on the difficult T2 test dataset challenge the common assumption that a domain-specific FM will outperform a broader FM.
>
> **Regarding our choice of total accuracy as a metric**: this choice was deliberate for our experiments as in the case of view recognition no errors are more costly than others (which could be the case if classes were different types of diagnoses). Instead, the total number of errors and thereby the potential number of corrections needed by the clinician is the most important factor, and the distribution of the samples in our training-and test-set corresponds to the typical situation in a clinic. Per-class accuracies have however been studied. Here, we observed that the difficulty of the classes has the largest effect. Some views look very similar to each other (for example 4CH and 5CH, especially when the image is acquired during the transition from one view to the other) and can be confused by the model.  A small class like NO-ORGAN is an easy class and often has the best performance of all.
>
> **Regarding our hypothesis for DINOv2’s superior performance**: We assume the diversity and the size of the pre-training dataset (142 million curated images) is the main reason why DINOv2 outperforms other general foundation models. Additional factors could be the use of aggressive augmentations and model distillation in the DINOv2 training regime. It has also been shown that contrastive models (like EchoCLIP and MedImage Insight) typically have lower fine-tuning performance than masked image models like DINOv2 [SSL Cookbook section 3.7.1]

---

### Meta-Review · Area_Chair_aBcC · 2025-10-29

**Recommendation:** Accept (Poster)
**Confidence:** 4

**Metareview:**

Although this work has some weaknesses - such as limited novelty and evaluation - its strengths, including the significance of its results and findings, appear to outweigh these limitations. Overall, I would recommend accepting this paper.

---

### Decision · Program_Chairs · 2025-11-05

**Decision:**

Accept (Poster)

**Comment:**

We recommend a poster presentation given the AC and reviewers recommendations.